# The Effect of Health Shock and Basic Medical Insurance on Family Educational Investment for Children in China

**DOI:** 10.3390/ijerph18105242

**Published:** 2021-05-14

**Authors:** Pu Liao, Zhihong Dou, Xingxing Guo

**Affiliations:** China Institute of Actuarial Research, School of Insurance, Central University of Finance and Economics, Beijing 102206, China; 2018110066@email.cufe.edu.cn (Z.D.); gxxleiyan@163.com (X.G.)

**Keywords:** health shock, basic medical insurance, investment in child education, OLG

## Abstract

This paper explores the role of basic medical insurance in protecting family investment in child education. First, this paper establishes a two-phase overlapping generation model to theoretically analyse the impact of basic medical insurance on investment in child education under the influence of the impact of parental health. The results show that health shock reduces parental investment in child education, and medical insurance significantly alleviates the negative impact of parental health shock on investment in child education. Furthermore, this paper establishes a two-way fixed effect regression model based on the data of China Family Panel Studies (CFPS) in 2014 and 2016 to empirically test the above results. The results showed that parental health shocks negatively affect investment in child education, and paternal health shock has a more significant impact than maternal health shock. However, medical insurance significantly reduces this negative impact, provides security in investment in child education, and promotes the improvement of human capital.

## 1. Introduction and Literature Review

Human capital is an important driving force of economic development. Education investment improves the level of human capital and promotes the development of productivity. Although government expenditure plays a very important role in the field of education, families remain the main investors and executors of education, and family education expenditure is one of the major expenditures among families. According to the data from the China Household Finance Survey (CHFS) by the Southwest University of Finance and Economics, in the second semester of 2016 and the first semester of 2017, the overall scale of family education expenditure on pre-school and basic education was approximately 1904.26 billion yuan, accounting for 2.48% of the GDP in 2016, and the total amount was equivalent to 60% of financial education funds. The average family education expenditure on pre-school and primary and secondary school students was 8143 yuan, accounting for 13.2% of the total household consumption expenditure.

Health risk is one of the most important and common risks faced by family members. Health shock increases family medical expenditure [1,2,3], reduces family labor income [4,5,6] and subsequently changes family economic decisions. Because the family is the main undertaker of education investment, and family education expenditure is one of the major expenditures of families, health impacts are likely to affect family education investment. According to China Health Statistical Yearbook 2019, the 2-week prevalence rate was 24.1%, and the prevalence rate of chronic diseases was 22.7%. According to China Family Panel Studies 2018 (CFPS2018) published by Peking University, the per capita disposable income of urban residents was 22,201 yuan; 17.8% of the individuals hospitalized for diseases had medical expenses exceeding that amount; and 9.37% of residents had out-of-pocket medical expenses above 22,201 yuan. On the basis of Chinese data, Reference [7] believes that health shock will result in a price effect and consequently change family preferences; their research shows that when parents experience health shock, their children are 9.9% less likely to enter junior high school.

Medical insurance is generally accepted to alleviate the adverse economic impact of health shock (see [5,6]). Therefore, China government places great importance on the construction of a medical security system. In 1998, 2003 and 2007, China began to establish a basic medical insurance system for urban workers, a new rural cooperative medical insurance system and a medical insurance system for urban residents, respectively, to achieve coverage for all people. According to the China National Health Insurance Bureau, the basic medical insurance coverage rate reached 95% in 2019.

In summary, family child education expenditure may be decreased by parental health shock, and basic medical insurance alleviates the economic impact of health shock. Would basic medical insurance affect family education expenditure under health shock?

Many researchers have studied these issues. Some have studied the relationship between health shock and family education decision-making. Reference [8] has found that if children are considered substitutes in the labor market or are required to work at home (family affairs), the long-term health impact on parents may affect child education. Reference [7] has found that the health shock of adults negatively affects the enrollment rate of children: when the adults in the family are seriously ill, the chance of children in primary school entering secondary school is reduced by 9.9 percentage points, while Reference [9] has found that children living with sick mothers are much less likely to receive education in the 15–24 year age group. Reference [10] has found that only paternal health shock has a negative impact on school attendance, whereas health shock in the mother or other family members has no such impact. Reference [11] has investigated the relationship between parental illness and children’s engagement in education and the labor market, using panel data from Vietnam. Their findings have shown that maternal illness substantially decreases the chances of children 11–23 years old being enrolled in school and simultaneously increases children’s likelihood of entering the labor market and working more hours. This effect is particularly pronounced for girls. Using two waves (2007, 2014) of Indonesian Family Life Survey data, Reference [12] has investigated the effects of parental chronic illness on the educational attainment of children in Indonesia. Their results show that girls whose fathers experienced chronic illness for longer durations achieved significantly lower educational levels between 2007 and 2014 than did children in the same age cohort with healthy parents. In contrast, boys were unaffected by the duration of the father’s chronic illness.

Other scholars have studied the effects of public health insurance on family education under health shock. Reference [13] has found that the health impact of the head of household or spouse has a significant negative effect on the school enrollment rate of rural children in China, and the new rural cooperative medical insurance has played a role in alleviating this effect. On the basis of family survey data from Rwanda, Reference [14] has found that parental health insurance significantly decreases the negative impact of health shock on the child school enrollment rate. Reference [15] has found that the expansion of Mexico’s public health insurance program has had a significant positive impact on child school enrollment and academic performance. Reference [16] has found that public health insurance improves the ability of American child immigrants to undergo and pay more for education. Using national health insurance data in Ghana, Reference [17] has shown that insurance programs reduce the risk of health shock causing families to take their children out of school so that they can go to work. Using two rounds of nationally representative survey data, Reference [18] has found that India’s national health insurance scheme is beneficial for child education: school expenditure increased as a result of the treatment. The existing literature has conducted in-depth research on the interactions among health shock, medical insurance, and family education decision-making. With respect to the existing literature, this paper makes two contributions: First, the existing literature [8,9,10,11,12,13,14,15,16,17,18] has used the enrollment rate as a measure of family education decision-making, but given that China’s primary and junior high school enrollment rate is close to 100% (according to the statistical monitoring report of China’s children development program (2011–2020) issued by the National Bureau of Statistics in 2019, the net enrollment rate of primary school-age children was 99.95%, the gross enrollment rate in junior middle school was 100.9%, and the population coverage rate in 9-year compulsory education reached 100%), this variable will no longer be applicable in China. With a theoretical model and empirical model, this paper uses education expenditure as a measure of family education decision to improve research in this field. Second, the research [8,13] on China in this field has mainly focused on Chinese rural families; this paper expands the research to all families to draw more general conclusions.

In this paper, we first establish a two-stage overlapping generation (OLG) model to study the optimal educational investment decisions of insured families and uninsured families when health shock occurs or does not occur, and we analyze the impact of basic medical insurance on child educational expenditure under health shock. According to the theoretical results, on the basis of the data from China Family Panel Studies 2014 and 2016 (CFPS), this paper establishes an econometric model to analyze the impact of parental health shock on child education expenditure from micro perspective and assesses whether medical insurance alleviates the adverse impact of health shock on child education expenditure. This study shows that parental health shock negatively affects family investment in child education, and basic medical insurance significantly decreases this negative impact.

## 2. Theoretical Model

In this paper, a two-stage OLG model is established to analyze family education behavior (this paper refers to [14] and establishes a theoretical model to get a reasonable hypothesis and research expectations before empirical test. It is natural to employ OLG model because we focus on the parental educational decisions for their children). Suppose that there are many families in the economy, and each family includes two generations of parents and children. Parents invest in the education of their growing children during the working period and receive part of the income from their children’s wages during the retirement period.

Let $w_{t}$ stand for the fixed wage rate and *d_t_* stand for the working hours. We assume that the child’s salary $\left(w_{t+1}\right)$ is determined by the parental salary $\left(d_{t}w_{t}\right)$ and education investment *e_t_*, on the basis of the Cobb Douglas production function to describe the child wage growth model, referring to [19]; that is:(1)$$w_{t+1}=Ae_{t}^{\delta }{\left(d_{t}w_{t}\right)}^{1-\delta }$$
where *A* and *δ* represent the efficiency of education expenditure to improve the child’s wage level.

Assume that if the working time *dt* of parents in a healthy state is 1, the disease will reduce the parental working time; that is, if the parents become ill in the working period, then their working time ($d_{t}^{x,y}$) will be less than 1. Referring to [20], we assume that the working time under the condition of disease can be expressed as:(2)$$d_{t}^{x,y}=1-hf(m_{t}^{x,y})$$
(3)$$f(m_{t}^{x,y})={\left(1+m_{t}^{x,y}\right)}^{-1}$$
where *x* represents the situation in which parents have medical insurance (*x* = *u* means uninsured, *x* = *i* means insured); y represents the health status of parents (*y* = *n* means not sick, *y* = *s* means sick); *h*(*h* ≥ 0) is the exogenous fixed time cost caused by health shock; and *f*(.) is the proportion of working time loss caused by health shock after paying medical expenses $\left(m_{t}^{x,y}\right)$, and $f^{\prime }(.)<0,\;f^{\prime \prime }(.)>0,\;0\leq f(.)\leq 1$. 

The above formula indicates that when parents experience health shock, their working hours will decrease *h*, but parents can choose to spend medical expenses $m_{t}^{x,y}$ to reduce the impact of health shock on their working hours. Notably, when *f*(.) = 1 and *h* = 1, parents do not work; when *h* = 0, health shock has no effect on parental working hours, or no health shock; when *f*(.) = 0, medical expenditure successfully eliminates the effect of health shock on parental working hours.

Suppose that there is an insurance market, parents decide whether to buy medical insurance at the beginning. If parents buy medical insurance in period *t*, the premium expense is *p*, and if they become sick during the insurance period (in period *t*), they receive compensation of ${\gamma m}_{t}^{i,s}$, where 0≤γ≤1 represents the compensation proportion. Reference [21] has indicated that after the purchase of medical insurance, owing to the decrease in the proportion of out-of-pocket medical costs, the willingness of individuals to spend on medical expenses increases, and the total medical expenses of insured parents exceed those of uninsured parents. This paper referred to [20] to define medical expenditure:(4)$$m_{t}^{u,s}=\;\sqrt{hw_{t}}-1$$
(5)$$m_{t}^{i,s}=\;\sqrt{\frac{hw_{t}}{\left(1-\gamma \right)}}-1$$

Therefore, families can be divided into four categories according to whether the parents buy insurance and experience health shock.

In the first category, the parents did not buy insurance and did not encounter health shock. The salary of this type of family in phase *t* is *w_t_*, and because there is no need to pay medical expenses, the budget constraint is:(6)$$c_{t}^{u,n}=w_{t}-s_{t}^{u,n}-e_{t}^{u,n}$$
(7)$$c_{t+1}^{u,n}=s_{t}^{u,n}(1+r)+\alpha (A{\left(e_{t}^{u,n}\right)}^{\delta }w_{t}^{1-\delta })$$
where $c_{t}^{x,y}$ represents the consumption level of parents in period *t* when the health status is *x*, and the medical insurance status is *y*; $s_{t}^{x,y}$ represents the family’s savings in period *t* when the health status is *x*, and the medical insurance status is *y*; *r* is the rate of return of savings investment; and *α* is dependency ratio (the proportion of children’s wage income transferred to their parents), α>0.

In the second category, the parents did not buy insurance and had health shock. The salary of this type of family in period t is $d_{t}^{u,s}w_{t}$, and medical expenses $m_{t}^{u,s}$ must be paid, so the budget constraint is:(8)$$c_{t}^{u,s}=d_{t}^{u,s}w_{t}-s_{t}^{u,s}-e_{t}^{u,s}-m_{t}^{u,s}$$
(9)$$c_{t+1}^{u,s}=s_{t}^{u,s}(1+r)+\alpha (A{\left(e_{t}^{u,s}\right)}^{\delta }{\left(d_{t}^{u,s}w_{t}\right)}^{1-\delta })$$

In the third category, the parents purchased insurance and did not experience health shock. The salary of this type of family in period t is $w_{t}$, and the premium *p* must be paid, so the budget constraint is:(10)$$c_{t}^{i,n}=w_{t}-s_{t}^{i,n}-e_{t}^{i,n}-p$$
(11)$$c_{t+1}^{i,n}=s_{t}^{i,n}(1+r)+\alpha (A{\left(e_{t}^{i,n}\right)}^{\delta }w_{t}{}^{1-\delta })$$

In the fourth category, the parents purchased insurance and encountered health shock. The salary of this type of family in period t is $d_{t}^{u,s}w_{t}$, and the premium *p* and self-paid medical expenses $(1-\gamma )m_{t}^{i,s}$ must be paid; therefore, the budget constraint is:(12)$$c_{t}^{i,s}=d_{t}^{i,s}w_{t}-s_{t}^{i,s}-e_{t}^{i,s}-(1-\gamma )m_{t}^{i,s}-p$$
(13)$$c_{t+1}^{i,s}=s_{t}^{i,s}(1+r)+\alpha (A{\left(e_{t}^{i,s}\right)}^{\delta }{\left(d_{t}^{i,s}w_{t}\right)}^{1-\delta })$$

The utility function of parents is defined as:(14)$$\underset{s_{t},e_{t}}{\max}U^{x,y}=\ln(c_{t}^{x,y})+\beta \ln(c_{t+1}^{x,y})$$
where *β* is the utility discount factor.

In the above constraints, savings and education investment are decision variables, and the optimal education investment decision can be obtained by the first-order condition when the utility is maximized:(15)$$e_{t}^{u,n}={\left(\frac{1+r}{\alpha A\delta }\right)}^{\frac{1}{\delta -1}}w_{t}$$
(16)$$e_{t}^{u,s}={\left(\frac{1+r}{\alpha A\delta }\right)}^{\frac{1}{\delta -1}}d_{t}^{u,s}w_{t}$$
(17)$$e_{t}^{i,n}={\left(\frac{1+r}{\alpha A\delta }\right)}^{\frac{1}{\delta -1}}w_{t}$$
(18)$$e_{t}^{i,s}={\left(\frac{1+r}{\alpha A\delta }\right)}^{\frac{1}{\delta -1}}d_{t}^{i,s}w_{t}$$

According to Equations (15)–(18), we reach the following conclusions:

*Conclusion 1*: The child education investment of families with health shock is less than that of healthy families, thus indicating that health shock reduces parental investment in child education.

*Proof.* Because $d_{t}^{x,y}\leq 1$, therefore $e_{t}^{u,s}\leq e_{t}^{u,n}$, $e_{t}^{i,s}\leq e_{t}^{i,n}$. □

*Conclusion 2*: The amount of education expenditure reduced by health shock in insured families is less than that in uninsured families; therefore, medical insurance weakens the negative impact of health shock on investment in child education.

*Proof*. From formulas (3) and (4), we obtain $m_{t}^{u,s}<m_{t}^{i,s}$,

From formulas (1) and (2), it is concluded that $d_{t}^{u,s}<d_{t}^{i,s}$,

so $e_{t}^{u,s}\leq e_{t}^{i,s}$, and because $e_{t}^{u,n}=e_{t}^{i,n}$, then $e_{t}^{u,n}-e_{t}^{u,s}>e_{t}^{i,n}-e_{t}^{i,s}$. □ 

According to the above conclusions, we can formulate hypothesis that must be further verified with empirical tests:

**Hypothesis** **(H1).**
*Controlling for other factors, parental health shock will negatively affect educational investment of children.*


**Hypothesis** **(H2).**
*Controlling for other factors, parental participation in basic medical insurance will alleviate the adverse effects of health shock and play a role in protecting the family’s investment in children’s education.*


## 3. Regression Model Setting, Index Selection, and Data Description

### 3.1. Regression Model Setting

Referring to [13], this paper uses a fixed effect regression model to test the impact of parental health shock on investment in child education level:(19)$$\ln Y_{ijt}=\lambda_{1}H_{ijt}^{f}+\lambda_{2}H_{ijt}^{m}+\beta X_{ijt}+\gamma_{j}+\gamma_{t}+c_{i}+\varepsilon_{ijt}$$
where *i* refers to the family; *j* refers to the area where the family is located, which is divided into eastern, central, western, and northeastern regions; *t* refers to the year; ln*Y_ijt_* is the level of investment in child education, which is measured by the logarithm of child total education expenditure in the past year; $H_{ijt}^{f}$, $H_{ijt}^{m}$ are the health status of the child’s father and mother, respectively, that is, whether the father or mother has had health shock, where health shock is 1 and no health shock is 0; the coefficient $\lambda_{1}$, $\lambda_{2}$ represent the influence of the father’s and mother’s health shock on investment in the child education level, respectively, $X_{ijt}$ is the characteristic variable of family i, including the child’s gender, child’s age, whether the residence is urban, whether the child lives with the mother, the mother’s education level, whether the parents have endowment insurance, the logarithm of family per capita income, and the number of children in the family; the coefficient *β* represents the influence of other characteristic variables in the family on the level of investment in child education; $\gamma_{j}$ is a fixed effect in the eastern, central, and western regions; $\gamma_{t}$ is a dummy variable in the survey year; *c_i_* is an individual fixed effect; and $\varepsilon_{ijt}$ is a random disturbance term that contains other information other than the variables used in the model.

In the above model, the coefficient λ measures the impact of health shock on the education investment level. If families alleviate or eliminate health shock through insurance or other means, the value of λ will tend to 0. To analyze whether the insured behavior might alleviate or eliminate the impact of health shock on the level of family education investment, this paper refers to [13] and extends the coefficient $\lambda_{1}$, $\lambda_{2}$ to Equations (20) and (21):(20)$$\lambda_{1}=\alpha_{0}+\alpha_{1}R_{ijt}^{f}+\gamma_{j}+\gamma_{t}+v_{ijt}$$
(21)$$\lambda_{2}=\alpha_{0}^{\prime }+\alpha_{1}^{\prime }R_{ijt}^{m}+\gamma_{j}+\gamma_{t}+v_{ijt}$$
where $R_{ijt}^{f}$, $R_{ijt}^{m}$ are the dummy variable of whether the father or mother of the *i*-th family in area j is insured in year t; insurance is 1 or otherwise is 0; $\alpha_{1}$ measures the influence of father’s participation on $\lambda_{1}$; $\alpha_{1}^{\prime }$ measures the influence of the mother’s participation on $\lambda_{2}$; and $\gamma_{t}$ is the change in $\lambda_{1}$ (or $\lambda_{2}$) caused only by the change in time, which may not be related to insurance. Regional fixed effect $\gamma_{j}$ measures the difference of (or $\lambda_{2}$) caused by regional heterogeneity. Substituting Equations (20) and (21) into Equation (19) yields the following:(22)$$\ln Y_{ijt}=\alpha_{0}H_{ijt}^{f}+\alpha_{0}^{\prime }H_{ijt}^{m}+\alpha_{1}H_{ijt}^{f}\times R_{ijt}^{f}+\alpha_{1}^{\prime }H_{ijt}^{m}\times R_{ijt}^{m}+\gamma_{t}H_{ijt}^{f}+\gamma_{t}H_{ijt}^{m}+\gamma_{j}H_{ijt}^{f}+\gamma_{j}H_{ijt}^{m}+\beta X_{ijt}+\gamma_{j}+\gamma_{t}+c_{i}+\varepsilon_{ijt}$$
in which when $R_{ijt}^{f}=0$; that is, when the father did not participate in basic medical insurance, the coefficient of $H_{ijt}^{f}$ is $\alpha_{0}$; when $R_{ijt}^{f}=1$, that is, when the father participated in basic medical insurance, the coefficient of $H_{ijt}^{f}$ is $\alpha_{0}+\alpha_{1}$; the coefficient of $H_{ijt}^{m}$ is the same. Therefore, the coefficients $\alpha_{0}$ and $\alpha_{0}^{\prime }$ measure the direct impact of health shock of an uninsured father or mother on ln*Y_ijt_*; $\alpha_{0}+\alpha_{1}$ and $\alpha_{0}^{\prime }+\alpha_{1}^{\prime }$ measure the total effect of health shock on the insured father or mother; the coefficients $\alpha_{1}$ and $\alpha_{1}^{\prime }$ of the interaction term $\left(H_{ijt}\times R_{ijt}\right)$ between the insured parents and health shock measure the difference in the impact of parental health shock on family education investment between insured and uninsured families, and determine the role of medical insurance in reducing the adverse impact of health shock on family education investment; the coefficient of the interaction term of $\gamma_{t}$ and $H_{ijt}$ represents the change in the impact of health shock on the dependent variable due to the change in time; and the interaction term of $\gamma_{t}$ and $H_{ijt}$ controls the continuous difference in the severity of health shock in different regions.

### 3.2. Data Sources and Descriptions

#### 3.2.1. Data Sources

The data used in this paper are from the China Family Panel Studies (CFPS) of the China Social Science Survey Center of Peking University in 2014 and 2016. Because the survey was late onset, the sample data from the survey are actually the observational data for 2013 and 2015. This paper examines 6–15-year-old children and their parents. After eliminating some missing data, we obtained 5090 samples, covering 28 provinces, municipalities, and autonomous regions.

#### 3.2.2. Data Selection and Propensity Score Matching (PSM)

To observe whether medical insurance might alleviate the adverse effects of health shock, the tracking data of the same individual before and after insurance should theoretically be compared; however, we had only the data for different individuals before and after insurance, which would have led to sample selection bias. To eliminate sample selection bias, we used the propensity matching method proposed by [22] to select data, that is, to calculate the probability *p*(*x*) of parental participation in basic medical insurance given some pre-treatment characteristics (*X_ijt_*), as shown in Equation (23)
(23)$$p(x)=\mathrm{Pr}[R_{ijt}=1|X_{ijt}]$$

In this paper, the propensity score matching method was used to process the data for 5090 samples. The first step was to estimate the propensity score function $\mathrm{Pr}[R_{ijt}=1|X_{ijt}]$, that is, the probability of parents participating in basic medical insurance given a set of observable characteristics. The logit model was used to estimate the propensity score of each individual, and then the children in the sample were matched. The factors selected in this paper were parental age, parental education level, family per capita income, family medical expenses, and whether the residence was urban. In the second step, we used the most common matching strategy, the k-nearest neighbor matching method in caliper, selecting a caliper distance of 0.008 (caliper distance is the standard deviation of tendency score multiplied by 0.25; that is, 0.00866. Therefore, the caliper range was set to 0.008), and carried out one-to-four matching for the observation values with an 0.8% difference in propensity score. The paired samples passed the co-support and matching test (see [23]). Finally, 5058 samples were obtained after elimination of the samples that did not support matching.

We performed empirical analysis of 5090 samples before propensity score matching and 5058 samples after propensity score matching to observe the impact of health on investment in child education level.

### 3.3. Variable Selection and Descriptive Statisticss

In this paper, the explained variable was the level of educational investment for the family’s children, and the index used was the logarithm of the total child educational expenditure in the past year of the survey year, including school education expenditure, extra-curricular guidance fees, and the cost of purchasing teaching materials. The explanatory variable was whether parents were affected by health shock. According to the results of the questionnaire survey, this paper regarded self-rated unhealthy and general states as indicating health shock, and relatively healthy, healthy, and very healthy states as indicating an absence of health shock. According to the explanation of the China National Bureau of statistics, the study area was divided into four regions: western, northeastern, central, eastern. The selection and implication of variables are shown in Table 1.

To observe the impact of health shock on family per capita income and family education investment, we divided all samples into two groups according to whether the family participated in basic medical insurance: the families with both parents participating in basic medical insurance accounted for 87.43%. Others were uninsured families, accounting for approximately 12.57%. The characteristics of the sample data are shown in Table 2.

The comparison shows that the proportion of parents with health shock and diagnosed chronic diseases in the two families is approximately the same; the average child education expenditure, family medical expenditure, and per capita family income of uninsured families are less than those of insured families. This finding shows that the disease treatment effect on uninsured families may be worse than that on insured families, and the working hours of parents may be adversely affected for longer times, which may cause the income of uninsured families and the investment in child education to be more negatively affected by health shock, whereas the medical insurance of insured families would alleviate the negative impact of health shock. However, the above explanation requires further empirical verification. In addition, the participation rate in parental endowment insurance of uninsured families was also lower than that of insured families, thus indicating that the insurance level of uninsured families was low.

## 4. Empirical Results and Analysis

### 4.1. Results and Analysis of Population Sample Estimation

First, according to the econometric model in Equation (22), we regressed the the matched sample. The results are shown in Table 3.

According to Table 3, the estimated values of $\alpha_{0}$ and $\alpha_{0}^{\prime }$ are negative (however, the results show that the impact is not significant, that is to say, the family’s education expenditure on children has a certain degree of rigidity), thus indicating that both paternal and maternal health shock negatively affect investment in child education expenditure. Coefficient $\alpha_{0}$ shows that the family’s expenditure on child education decreases by approximately 47.27% (suppose that the education expenditure of a family without health shock is $y_{0}$, and that of a family with the father experiencing health shock and without insurance is $y_{1}$, $\ln y_{1}-\ln y_{0}=-0.64$, that is, $y_{1}/y_{0}=e^{-0.64}$ ≈ 52.73%). In contrast, the estimated values of $\alpha_{1}$ and $\alpha_{1}^{\prime }$ are both positive and are significant at the levels of 10% or 5%, thus indicating that the basic medical insurance system can increase the level of investment in child education by alleviating the negative impact of health shock, and paternal participation can prevent approximately 43.64% (it is further assumed that the family education expenditure when the father experiences health shock and participates in insurance is $y_{1}^{\prime }$. In the same way, $y_{1}^{\prime }/y_{0}=e^{\left(-0.64+0.603\right)}\approx$ 96.37%. 96.37%–52.73% = 43.64%) of the reduction in education investment. Finally, the estimated value of $\alpha_{0}+\alpha_{1}$ is negative; that is, the medical insurance system cannot completely eliminate the negative impact of paternal health shock (from the perspective of maternal health shock and the parameters of maternal insurance, the medical insurance system completely eliminates maternal health shock, and even invest more in education than before. This may be due to the health shock changing the risk preference of families and the substitution effect of the medical insurance system, please see [8]).

The conclusions are consistent between the theoretical and empirical results; that is, health shock reduces parental investment in child education, and medical insurance can alleviate the negative impact of health shock on investment in child education.

### 4.2. Subsample Estimation Results

In this section, the samples selected by PSM were grouped and regressed to explore whether differences exist in the impact of health shock and medical insurance on the education investment of family children for different groups. Our paper classified the groups according to the types of areas where they live (urban or rural), the per capita income level of the family, the educational level of the mother, and the gender of the children.

#### 4.2.1. Analysis of the Differences in the Impact of Health Shock and Medical Insurance between Urban and Rural Families

All sample families were divided into rural and urban groups according to the urban and rural categories of the child’s residence. There were 3041 rural samples and 2017 urban samples. The econometric model of Equation (22) was used to regress the two groups of samples to explore the impact of health shock and medical insurance on the level of investment in child education in urban and rural families. The results are shown in Table 4.

According to the grouping regression results in Table 4, rural investment in child education is significantly negatively affected by paternal health. The absolute value in the rural group was 0.643, and the absolute value in the rural group was 0.647, thus indicating that the alleviating effect of rural paternal participation in insurance significantly reduced the negative impact of health shock. The results showed that when families experienced health shock, rural families were more willing to sacrifice their child education expenditures to alleviate the adverse effects, whereas urban residents paid more attention to child education and did not significantly reduce their child education expenditure. Therefore, the relief effect of medical insurance is more effective for rural families and can significantly reduce or even eliminate the negative impact of parental health on investment in child education.

#### 4.2.2. Difference Analysis of Health Shock and Medical Insurance on Families with Different Incomes

All samples were divided into three levels according to the family per capita income. Of the total number of people, the top 33% were high-income, 33% to 66% were middle-income, and the rest were low-income. According to the econometric model of Equation (22), the samples were grouped and regressed to explore the difference in the impact of health shock and medical insurance on the level of investment in child education in families with different income. The results are shown in Table 5.

Paternal health shock had a significant negative impact on the investment in child education in low-income families, whereas the investment in child education among middle- and high-income people affected by health shock was not significant. In addition, in the low-income group, the absolute value of the paternal health shock coefficient was larger than that in the middle and high-income groups, and the interaction coefficient between parental health shock and insurance was more significant. The reason for this result may be that low-income people are more vulnerable to health shock, and consequently the proportion of negative impact was higher. In addition, for the high-income group, the alleviating effect of health insurance was not significant, because the investment in education tended not to change, even in the presence of health impacts, high-income families gave priority to maintaining education expenditure.

#### 4.2.3. Difference Analysis of Effects of Health Shock and Medical Insurance on Mothers with Different Education Levels

To explore the impact of health shock and medical insurance on investment in child education level among families with different maternal educational levels, we divided the samples into two groups according to the maternal educational level: mothers with primary school education or below, and mothers with primary school education or above. There were 2228 samples of mothers with primary school education or below and 2830 samples of mothers with primary school education or above. According to the econometric model of Equation (22), the samples were grouped and regressed, and the results are shown in Table 6.

In the families in which the mother had primary school education or below, the negative impact of paternal or maternal health shock on investment in child education was higher, and the mitigation effect of the mother’s medical insurance was more significant. The reason for this result may be that parents with lower education levels paid less attention to their investment in child education. When families encountered health shock, parents were more willing to sacrifice their child education expenditure to alleviate the adverse effects, whereas parents with higher education levels paid more attention to their child education and did not significantly reduce their child education expenditure, thus also indicating that education has relatively strong intergenerational transmission. Therefore, the protective effect of medical insurance on investment in child education is also greater for families with low parental education levels and can significantly reduce the negative impact of parental health on investment in child education.

#### 4.2.4. Impact of Health Shock and Medical Insurance on Educational Investment in Children of Different Genders

The samples were divided into boys and girls, including 2369 girls and 2689 boys. According to the econometric model of Equation (22), the two groups of samples were regressed to explore the impact of health shock and medical insurance on the child education investment level for children of different genders. The results are shown in Table 7.

Comparison of the regression results indicated that, under parental health shock, the education expenditure on boys was lower than that on girls. The reason for this result may be that in ordinary families, boys are regarded in terms of their future productivity; they receive more investment in education, and when adverse shocks occur, they are more affected.

### 4.3. Robustness Test

To verify the reliability of the empirical results, we divided the CFPS data in 2014 into two groups: insured families and uninsured families. We then performed ordinary OLS estimation to test the robustness of the impact of basic medical insurance on investment in child education under health shock. The econometric models used are as follows:(24)$$\ln(Y_{i})=\alpha_{0}+\alpha_{1}H_{i}^{f}+\alpha_{2}H_{i}^{m}+\alpha_{3}C_{i}^{f}+\alpha_{4}C_{i}^{m}+X_{i}\beta^{\sim k}+\varepsilon_{i}\;(\mathrm{uninsured})$$
(25)$$\ln(Y_{i})=\alpha_{0}^{\prime }+\alpha_{1}^{\prime }H_{i}^{f}+\alpha_{2}^{\prime }H_{i}^{m}+\alpha_{3}^{\prime }C_{i}^{f}+\alpha_{4}^{\prime }C_{i}^{m}+X_{i}{\beta^{\prime }}^{\sim k}+\varepsilon_{i}\;(\mathrm{insured})$$
where $\ln(Y_{i})$ is the logarithm of child total education expenditure in the past year, which represents the level of investment in child education, and $H_{i}^{f}$ and $H_{i}^{m}$ indicate whether the father or mother, respectively, has had health shock. If the self-rated health status is unhealthy or health shock is generally regarded as present, the value is 1; if the self-rated health status is relatively healthy, healthy, or very healthy, health shock is regarded to be absent, the value is 0. The coefficients $\alpha_{1}$ and $\alpha_{2}$ indicate the influence of paternal and maternal health shock on investment in the child education level, respectively; $C_{i}^{f}$ and $C_{i}^{m}$ indicate whether the father and the mother have chronic diseases, respectively: 1 indicates presence of chronic diseases, and 0 indicates absence of chronic diseases; $\alpha_{3}$ and $\alpha_{4}$ indicate the influence of paternal and maternal chronic diseases on investment in child education level, respectively. The control variables $X_{i}$ of family characteristics include the child’s gender, child’s age, whether the place of residence is urban, whether the child lives with the mother, the mother’s education level, whether the parents have endowment insurance, the logarithm of family per capita income, and the number of children in the family. The econometric models of Equations (23) and (24) were used for regression, and the regression results are shown in Table 8.

Comparison of the results of the two groups indicated that paternal health shock in uninsured families has a significant negative impact on investment in child education, and maternal health shock and chronic diseases also have a negative impact, whereas parental health shock and chronic diseases in insured families have no significant negative impact. Our findings thus indicate that medical insurance weakens the negative impact of health shock on investment in child education and provides support for the family. The level of investment in child education plays a protective role, in agreement with the results of the whole sample fixed effect model above, thus indicating that the regression results of this paper are reliable.

## 5. Conclusions

This paper constructed a two-stage OLG model including family education expenditure decision, studied the impact of basic medical insurance on investment in child education under the impact of parental health, and performed empirical analysis of the theoretical conclusions based on the CFPS survey data in 2014 and 2016. Compared with existing research, the OLG framework established in this paper is more consistent with the characteristics of intergenerational education and enables empirical analysis of the situation in China.

We draw the following conclusions: paternal maternal health shock negatively affects investment in child education expenditure, whereas basic medical insurance can significantly and effectively reduce the negative impact of health shock on education investment and protect the level of family human capital investment. In addition, we examined the family’s urban and rural attributes, income level, maternal education level, and child gender. The grouping test results show that rural families, low-income families, families in which the mother has a low educational background and families with boys are more likely to reduce family education expenditure when they are affected by health shock, and insurance can alleviate the impact of health shock on these families. However, the mitigation effect of insurance in these families is not necessarily greater than that in other families. For example, the mitigation effect of insurance in middle-income families is greater, possibly because insurance not only alleviates the amount of family income and expenditure, but also affects family preferences.

The results of this study show that basic medical insurance can protect the investment in child education under the risk of health shock and promote improvements in the human capital level. Therefore, the government should further increase the participation rate in basic medical insurance and expand the scope and proportion of reimbursement to improve the level of medical insurance. Because health impact has a greater negative effect on education investment in rural families, low-income families, and families in which the mother has a low educational background, the government should strengthen the protection of these vulnerable groups, e.g., through introducing preferential policies such as reducing or exempting insurance premiums or increasing the proportion of reimbursement for serious illness.

## Figures and Tables

**Table 1 ijerph-18-05242-t001:** Variable explanation.

Variable Name	Interpretation of Meaning
Investment level in child education	The logarithm of the total child education expenditure within 1 year, including school education expenditure, extra-curricular guidance fees, and textbook purchase fees.
Health shock	If the father’s or mother’s self-rated health status was unhealthy or health shock was considered present, it was 1; if the self-rated health status was relatively healthy, healthy, or very healthy, health shock was considered absent, and it was 0.
Chronic disease	In the past year, if the father or mother had chronic disease, it was 1, and otherwise was 0.
Participation in basic medical insurance	If fathers or mothers participated in any free medical service programs, medical insurance programs for urban employees, medical insurance programs for urban residents, or new rural cooperative medical insurance, they were considered to participate in basic medical insurance, it was equal to 1 and otherwise was 0.
Urban or rural residence	It was 1 for urban and 0 for rural.
Gender	It was 1 for boy and 0 for girl.
Age of the child	The age of the sample ranged from 6 to 15 years.
Living with mother	For children who lived with their mothers for more than 6 months, it was 1 and otherwise was 0.
Maternal education level	Maternal number of years of education.
Per capita household income	Total household income/total household population in the past year.
Participation in endowment insurance	If father or mother with endowment insurance, it was equal to 1 and otherwise was 0.
Number of children in family	The number of children in the family.
Region	Eastern, central, western, or northeastern (The eastern region includes Beijing, Tianjin, Hebei, Shanghai, Jiangsu, Zhejiang, Fujian, Shandong, Guangdong, and Hainan; the central region includes Shanxi, Anhui, Jiangxi, Henan, Hubei, and Hunan; the western region includes inner Mongolia, Guangxi, Chongqing, Sichuan, Guizhou, Yunnan, Tibet, Shaanxi, Gansu, Qinghai, Ningxia, and Xinjiang; the northeast region includes Liaoning, Jilin, and Heilongjiang).

**Table 2 ijerph-18-05242-t002:** Descriptive statistics of variables (Table 2 is only a simple statistical description of the data, which tends to show the overall differences between the insured families and the uninsured families. Our empirical tests are based on all families, not separately on insured families and uninsured families).

Variable Name	Uninsured Families			Insured Families		
	Sample	Mean	Variance	Sample	Mean	Variance
Expenditure on child education	325	2502.81	3770.85	2264	2679.62	3470.23
Family medical expenditure	324	4504.89	11992.57	2248	4670.93	10876.69
Per capita household income	318	10602.84	10455.73	2204	10871.25	11999.47
Health shock in father	326	0.18	0.39	2268	0.18	0.39
Health shock in mother	326	0.25	0.44	2268	0.25	0.43
Chronic disease in father	326	0.13	0.34	2268	0.11	0.32
Chronic disease in mother	326	0.14	0.34	2268	0.12	0.33
Mother’s number of years of education	326	7.67	4.20	2268	7.04	4.13
Age of the child	326	9.03	2.31	2268	8.72	2.57
Whether the residence is urban	318	0.48	0.50	2236	0.38	0.48
Child’s gender	326	0.52	0.50	2268	0.53	0.50
Number of children in family	323	1.91	0.83	2253	2.06	0.89
Participation of father in endowment insurance	326	0.46	0.50	2268	0.72	0.45
Participation of mother in endowment insurance	326	0.41	0.49	2268	0.67	0.47

**Table 3 ijerph-18-05242-t003:** Impact of health shock and medical insurance on family investment in child education.

	Ln (Family Education Input)
Paternal health shock	−0.640
	(0.4)
Maternal health shock	−0.358
	(0.311)
Paternal health shock × insured	0.603 *
	(0.34)
Maternal health shock × insured	0.613 **
	(0.267)
Paternal health shock × year	0.078
	(0.157)
Maternal health shock × year	0.093
	(0.133)
Paternal health shock × western region	−0.265
	(0.28)
Paternal health shock × northeastern region	0.115
	(0.311)
Paternal health shock × central region	−0.161
	(0.247)
Maternal health shock × western region	−0.043
	(0.0264)
Maternal health shock × northeastern region	−0.232
	(0.343)
Maternal health shock × central region	−0.318
	(0.235)
Maternal education level	0.04
	(0.048)
Age of the child	0.175 *
	(0.102)
Whether the residence is urban	−0.46 *
	(0.274)
Whether the child lives with the mother	−0.313 ***
	(0.101)
Gender of child	0.493
	(0.313)
Household income	−0.021
	(0.069)
Whether the father participates in endowment insurance	0.023
	(0.098)
Whether the mother participates in endowment insurance	0.013
	(0.095)
Number of children	0.000
	(0.000)
Year	−0.487 **
	(0.216)
Western region	0.000
	(0.000)
Northeastern region	0.000
	(0.000)
Central region	−0.87
	(0.934)
Eastern region	−0.866
	(0.826)
_cons	5.859 ***
	(1.315)
$R^{2}$	0.016
F	1.853
N	5058

Note: (1) the explained variable is ln (family education input); (2) the standard deviation of heteroscedasticity robustness is in brackets; (3) *, **, and *** indicate significance levels of 10%, 5%, and 1%, respectively.

**Table 4 ijerph-18-05242-t004:** Impact of health shock and medical insurance on investment in child education in urban and rural areas.

	Rural	Urban
Paternal health shock	−0.643 *	−0.076
	(0.443)	(0.656)
Maternal health shock	−0.483	−0.441
	(0.463)	(0.468)
Paternal health shock × insured	0.647 *	0.054
	(0.383)	(0.525)
Maternal health shock × insured	0.486	1.028 ***
	(0.402)	(0.395)

Note: This is a simplified table of regression results; other control variables still exist in the regression, but only four regression estimators, $\alpha_{0}$ and $\alpha_{0}^{\prime }$, $\alpha_{1}$ and $\alpha_{1}^{\prime }$ are listed; standard deviation is shown in brackets; *** and * indicate significant correlation at 1% and 10% levels, respectively.

**Table 5 ijerph-18-05242-t005:** Impact of health shock and medical insurance on investment in child education level in families with different income.

	Low Income	Middle Income	High Income
Paternal health shock	−2.589 **	−1.141	0.105
	(1.089)	(0.944)	(1.18)
Maternal health shock	−0.169	0.547	−0.36
	(0.754)	(0.65)	(0.601)
Paternal health shock × insured	2.066 **	2.303 **	0.028
	(0.9)	(0.993)	(1.047)
Maternal health shock × insured	0.317	0.967 *	0.739
	(0.655)	(0.568)	(0.485)

Note: *, ** indicate significance levels of 10% and 5%, respectively.

**Table 6 ijerph-18-05242-t006:** Impact of health shock and medical insurance on investment in child education level among families with different maternal educational levels.

	Mother Graduated from Primary School or Below	Mother Graduated from Primary School or Above
Paternal health shock	−0.902 *	−0.195
	(0.528)	(0.572)
Maternal health shock	−0.795	0.243
	(0.525)	(0.399)
Paternal health shock × insured	0.628	0.315
	(0.51)	(0.44)
Maternal health shock × insured	1.292 ***	−0.009
	(0.443)	(0.354)

Note: ***, * indicate significant correlation at 1% and 10% levels, respectively.

**Table 7 ijerph-18-05242-t007:** Impact of health shock and medical insurance on child educational investment level for children of different genders.

	Boy	Girl
Paternal health shock	0.844 *	−0.483
	(0.508)	(0.64)
Maternal health shock	−0.549	−0.131
	(0.477)	(0.414)
Paternal health shock × insured	−0.542	0.705
	(0.432)	(0.561)
Maternal health shock × insured	0.640	0.679 **
	(0.419)	(0.331)

Note: *, ** indicate significance levels of 10% and 5%, respectively.

**Table 8 ijerph-18-05242-t008:** OLS regression results for different families.

	Uninsured	Insured
Paternal health shock ($\alpha_{1}$)	−0.705 *	0.060
	(0.420)	(0.120)
Maternal health shock ($\alpha_{2}$)	−0.255	0.165
	(0.340)	(0.110)
Paternal chronic disease ($\alpha_{3}$)	0.289	−0.157
	(0.500)	(0.140)
Maternal chronic disease ($\alpha_{4}$)	−0.353	0.133
	(0.424)	(0.140)

Note: * indicates significance levels of 10%.

## Data Availability

Publicly available datasets were analyzed in this study. This data can be found here: http://www.isss.pku.edu.cn/cfps/ (accessed on 11 March 2020).

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
