# Peer review of "The Effect of Health Shock and Basic Medical Insurance on Family Educational Investment for Children in China"

_ijerph, 2021, doi:10.3390/ijerph18105242_

Round 1
Reviewer 1 Report
The article presents multi-faceted analyzes of the effect of health shock and basic medical insurance on family educational investment for children in China. A lot of assumptions that, in other terms, could change the results of the research.
Advantages of article:
- the structure of the article described in the introduction,
- reference to other studies,
- preparation of models,
- statistical analysis.
Disadvantages:
- more literature could be referenced and more recent literature could be used.
Reviewer 2 Report
Thanks for the invitation to review this manuscript. The current manuscript seems far from reaching the standard of publication. I suggest a major revision of the current version, and this manuscipt would not be suggested acceptance unless the comments have been taken seriously.
(1) The introduction section is poorly written, which seems an enumeration of prior research and lacks inherent logic. What's the difference between this manuscript and prior studies? What new knowledge and informaiton has this manscuipt added? What the research contributions of this manucsript? What reserach question this study would like to address? What's the rationale and neccesity of using OLG model? Why it is appropriate? The authors should remove the so-called organization of the manuscript, which is useless and redundant.
(2) The theoretical model is inconsistent with the regression model, which makes them seem rather separated. The theoretical model shows that the education expenditure is affected by variables such as working time of parents (eq.15-18) which is absent in the regression model.
(3) In the regression model settting (eq(19)-eq(22)), the three-dimension subscripts show that the regression model should be spatial regression model. However, the empirical results show that it is just a very common fixed effect panel model. Please clarify the inconsistence.
(4) In table 2, what about the situation that just one parent has participated in insurance? Which group would this situation belongs to? Mann Witney U test should be conducted to check the difference in variables between the two groups.
(5) The single term of "participation in public health insurance" should not be left in the regression model (Table 3), while the interaction term is included.
(6) The categories of regions are classified as four dummy variables, as shown in table 3. How do they be constructed as interaction term "paternal/maternal health shock * region"?
(7) According to much relevant literature about insurance system in China, there is also a kind of social health insurance "urban and rural resident basic medical insurance" which is merged from the "urban resident basic medical insurance" and "new cooperative medical scheme". However, this is ignored in the regression.
(8) The conduction of PSM is rather abrupt, as it is not a policy analysis using method such as difference-in-difference. The sepration of pair-wise subsamples would not help improve the analysis. Such sepration is very strange in this context. If the authors really care about the adverse selection effect regarding the health insurance, you need to carefully propose a research design to address relevant endogeneity problems.
(9) In table 4, table 5, table 6, table 7, table 8, the covariates are currently omitted. Please also provide the regressions including them.
(10)The conclusion is too brief, please provide detailed discussion about the findings in the section 4.
Reviewer 3 Report
Overall impression:
In my opinion the paper presents an original contribution. The topic is interesting. The title is appropriate. The methods are sufficiently detailed. Study design is appropriate for the question. Data and results supported the conclusions.
The manuscript could benefit from careful proof-reading.
